



**Rapid measurement of Watson-Crick to Hoogsteen exchange in unlabeled**
**DNA duplexes using high-power SELOPE imino [1]H CEST**
Bei Liu[1], Atul Rangadurai[1], Honglue Shi[2], and Hashim M. Al-Hashimi*[1,2]
*1. Department of Biochemistry, Duke University School of Medicine, Durham, NC,*
*USA*
*2. Department of Chemistry, Duke University, Durham, NC, USA*
*Correspondence to: hashim.al.hashimi@duke.edu*



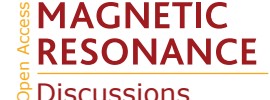

**Abstract.** In duplex DNA, Watson-Crick A-T and G-C base pairs (bps) exist in dynamic equilibrium with an alternative Hoogsteen conformation, which is low in abundance and short-lived. Measuring how the Hoogsteen dynamics varies across different DNA sequences, structural contexts and physiological conditions is key for understanding the role of these non-canonical bps in DNA recognition and repair. However, such studies are hampered by the need to prepare $^{13}$C or $^{15}$N isotopically enriched DNA samples for NMR relaxation dispersion (RD) experiments. Here, using SELective Optimized Proton Experiments (SELOPE) $^{1}$H CEST experiments employing high-power radiofrequency fields ($B_1 > 250$ Hz) targeting imino protons, we demonstrate accurate and robust characterization of Waston-Crick to Hoogsteen exchange, without the need for isotopic enrichment of the DNA. For 13 residues in three DNA duplexes under different temperature and pH conditions, the exchange parameters deduced from high-power imino $^{1}$H CEST were in very good agreement with counterparts measured using off-resonance $^{13}$C/$^{15}$N spin relaxation in the rotating frame ($R_{1\rho}$). It is shown that $^{1}$H-$^{1}$H NOE effects which typically introduce artifacts in $^{1}$H based measurements of chemical exchange can be effectively suppressed by selective excitation, provided that the relaxation delay is short ($\leq$100 ms). The $^{1}$H CEST experiment can be performed with ~10X higher throughput and ~100X lower cost relative to $^{13}$C/$^{15}$N $R_{1\rho}$, and enabled Hoogsteen chemical exchange measurements undetectable by $R_{1\rho}$. The



results reveal an increased propensity to form Hoogsteen bps near terminal ends
and a diminished propensity within A-tract motifs. The $^1$H CEST experiment opens
the door to more comprehensively characterizing Hoogsteen breathing in duplex
DNA.



# 1   Introduction

Soon after the discovery of the DNA double helix, it was shown that A-T and G-C
could also pair in an alternative conformation known as the "Hoogsteen" base pair
(bp) (Felsenfeld et al., 1957; Hoogsteen, 1959) (Fig. 1a).  Starting from a canonical
Watson-Crick G-C or A-T bp, the corresponding Hoogsteen bp can be obtained by
flipping the purine base 180º and bringing the two bases into proximity to create a
new set of hydrogen-bonds, which in the case of G-C bps require protonation of
cytosine-N3 (Fig. 1a).





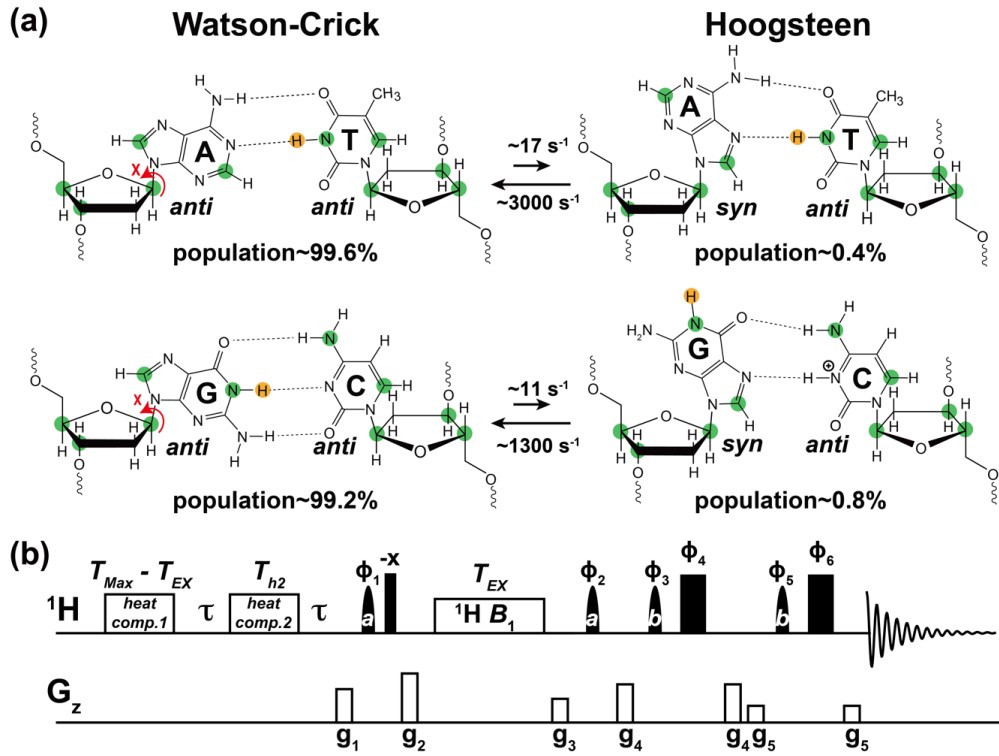

**Figure 1. Using ¹H CEST to measure Watson-Crick to Hoogsteen exchange**

**in unlabeled nucleic acid duplexes.** (a) Watson-Crick G-C and A-T bps in B-

DNA exist in dynamic equilibrium with G-C⁺ and A-T Hoogsteen bps, respectively.

Filled green circles denote nuclei (¹³C and ¹⁵N) that have previously been used to

probe the Watson-Crick to Hoogsteen exchange via RD measurements, while the

yellow circle denotes the imino ¹H probes used in this study. Rate constants and

populations were obtained as described previously (Alvey et al., 2014). (b) The

1D SELOPE ¹H CEST pulse sequence for characterizing chemical exchange in

unlabeled nucleic acids. Narrow and wide filled rectangles denote 90° and 180°



hard pulses. Semi-oval shapes denote selective pulses. Pulse **a** is a 90°
Eburp2.1000 shape pulse (typically 4 ms) for selective excitation of imino protons,
while pulse **b** is a 180° Squa100.1000 shape pulse with length 2 ms in an excitation
sculpting scheme (Hwang and Shaka, 1995) for water suppression. Open
rectangles denote the gradients and heat compensation elements. Delay $\tau$ = ½
$d_1$. To ensure uniform heating for experiments with variable lengths of $T_{EX}$, the
relaxation period during which a $^1$H $B_1$ field is applied, two heat compensation
modules were used according to a prior study (Schlagnitweit et al., 2018). The
first heat compensation is applied far off-resonance with duration = $T_{Max}$ - $T_{EX}$ = 2
ms, where $T_{Max}$ is the maximum relaxation delay time. The second heat
compensation (1 kHz) applied far off-resonance has a duration $T_{h2}$ = 150 ms. The
phase cycles used are $\phi_1$ = {8x, 8(-x)}, $\phi_2$ = {4x, 4(-x)}, $\phi_3$ = {x, y}, $\phi_4$ = {-x, -y}, $\phi_5$ =
{2x, 2y}, and $\phi_6$ = {2(-x), 2(-y)}. Gradients (g1 - g5) with SMSQ10.100 profiles are
applied for 1 ms with the following amplitudes (G cm$^{-1}$): 14.445, 26.215, 14.445,
16.585, 5.885. Briefly, imino $^1$H magnetization is selectively excited, aligned
longitudinally and then relaxes under a $^1$H $B_1$ field during $T_{EX}$. $^1$H transverse
magnetization is then created and directly detected following water suppression.
This pulse sequence is adapted from Schlagnitweit *et al* (Schlagnitweit et al.,

72   2018).






Following their discovery, Hoogsteen bps were observed in crystal structures of
duplex DNA in complex with proteins  (Kitayner et al., 2010; Aishima et al., 2002)
and drugs (Wang et al., 1984; Ughetto et al., 1985) and shown to play roles in DNA
recognition (Golovenko et al., 2018) , damage induction (Xu et al., 2020), and
repair (Lu et al., 2010), and in damage bypass during replication (Nair et al., 2006;
Ling  et  al.,  2003).    NMR  relaxation  dispersion  (RD)  studies  employing  off-
resonance $^{13}$C and $^{15}$N spin relaxation in the rotating frame ($R_{1\rho}$) later showed that
the  G-C  and  A-T  Watson-Crick  bps  exist  in  a  dynamic  equilibrium  with  their
Hoogsteen counterparts (Nikolova et al., 2011).  The Hoogsteen bps were shown
to be lowly populated (population < 1 %) and short-lived (lifetime ~ 1 ms) forming
robustly as an excited conformational state (ES) in duplex DNA across a variety of
sequence contexts (Alvey et al., 2014) (Fig. 1a).

There is growing interest in mapping the Watson-Crick to Hoogsteen exchange
landscape cross different DNA contexts, including for bps in different sequence
motifs (Alvey et al., 2014), near sites of damage and mismatches (Shi et al., 2021;
Singh et al., 1993), and when DNA is bound to proteins (Nikolova et al., 2013b;
Zhou et al., 2019) and drugs (Xu et al., 2018; Wang et al., 1984).  Studies suggest
an increased propensity to form Hoogsteen bps in such environments (Shi et al.,
2021) and this may in turn play roles in DNA recognition and damage repair (Afek



et al., 2020). Furthermore, there is interest in understanding how the Hoogsteen
exchange varies with temperature (Nikolova et al., 2011), pH (Nikolova et al.,
2013a), salt concentration and buffer composition (Rangadurai et al., 2020b;
Tateishi-Karimata et al., 2014), as well as in the presence of epigenetic
modifications (Wang et al., 2017; Rangadurai et al., 2019a), all of which could
shape these dynamics and consequently DNA biochemical transactions.

There are hundreds and thousands of motifs and conditions for which
characterization of Hoogsteen dynamics is of biological interest. However, current
approaches for measuring Hoogsteen dynamics are ill-suited for dynamics
measurements at such a scale. The Watson-Crick to Hoogsteen chemical
exchange process has been characterized with the use of $^{13}$C (Nikolova et al.,
2011; Shi et al., 2018; Ben Imeddourene et al., 2020; Alvey et al., 2014) and $^{15}$N
(Nikolova et al., 2012a; Rangadurai et al., 2019a; Alvey et al., 2014) off-resonance
$R_{1\rho}$, and more recently chemical exchange saturation transfer (CEST) experiments
(Rangadurai et al., 2020b; Rangadurai et al., 2020a). However, these approaches
require isotopically enriched DNA samples, making broad explorations of
Hoogsteen exchange across even tens of motifs impractical. Furthermore, many
motifs of interest involve damaged or modified nucleotides, which are difficult to
isotopically enrich with $^{13}$C and $^{15}$N nuclei. It is for this reason that we turned our





attention to the imino $^1$H as a probe of the Watson-Crick to Hoogsteen exchange
in unlabeled DNA samples.

The utility of protons as probes in CEST (Chen et al., 2016; Dubini et al., 2020;
Wang et al., 2021; Liu et al., 2020), Carr-Purcell-Meiboom-Gill (CPMG) (Juen et
al., 2016; Leblanc et al., 2018), and off-resonance $R_{1\rho}$ experiments (Wang and
Ikuta, 1989; Lane et al., 1993; Steiner et al., 2016; Schlagnitweit et al., 2018;
Baronti et al., 2020; Furukawa et al., 2021) to study conformational exchange in
nucleic acids is now well-established.  Many of these $^1$H based approaches use
experiments originally developed to study conformational exchange in proteins
(Ishima et al., 1998; Eichmuller and Skrynnikov, 2005; Lundstrom and Akke, 2005;
Lundstrom et al., 2009; Otten et al., 2010; Bouvignies and Kay, 2012; Hansen et
al., 2012; Weininger et al., 2012; Weininger et al., 2013; Smith et al., 2015; Sekhar
et al., 2016; Yuwen et al., 2017a; Yuwen et al., 2017b).  The $^1$H experiments permit
the use of higher effective fields allowing characterization of conformational
exchange faster than is possible using $^{13}$C or $^{15}$N experiments (Steiner et al., 2016;
Palmer, 2014).  Furthermore, the relationship between $^1$H chemical shifts and
structure is reasonably well understood and has been exploited in the
conformational characterization of nucleic acids (Sripakdeevong et al., 2014;



Frank et al., 2013; Wang et al., 2021; Swails et al., 2015; Czernek et al., 2000;
Lam and Chi, 2010).

Recently, $^1$H $R_{1\rho}$ and CEST SELective Optimized Proton Experiments (SELOPE)
were developed and applied to characterize conformational exchange in unlabeled
RNA (Schlagnitweit et al., 2018).  The SELOPE experiment has already found
several applications in studies of unlabeled nucleic acids, including in the
characterization of fast ($k_{ex} = k_1 + k_{-1} > 1,000$ s$^{-1}$) RNA secondary structural
rearrangements (Baronti et al., 2020) and DNA base opening (Furukawa et al.,
2021), as well as slower ($k_{ex} < 100$ s$^{-1}$) DNA hybridization kinetics (Dubini et al.,
2020).  Many $^1$H relaxation dispersion (RD) studies have targeted exchangeable
imino protons (Baronti et al., 2020; Furukawa et al., 2021), taking advantage of the
well-known dependence of the imino $^1$H chemical shifts on secondary structure
(Wang et al., 2021; Lam and Chi, 2010).

Although $^1$H RD experiments can obviate the need for isotopic labeling and offer
other advantages such as high sensitivity, they have not been as widely used
compared to $^{13}$C/$^{15}$N RD experiments.  One reason for this has to do with potential
artifacts arising due to from $^1$H-$^1$H cross relaxation (Ishima et al., 1998; Eichmuller
and Skrynnikov, 2005; Lundstrom and Akke, 2005; Bouvignies and Kay, 2012).

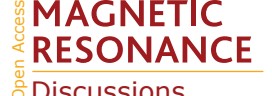

Interestingly, in nucleic acids, such NOE effects appear to be effectively
suppressed in the $^1$H SELOPE experiment through selective excitation of spins
(Schlagnitweit et al., 2018). The exchange parameters obtained using $^1$H SELOPE
experiments were shown to be in very good agreement with counterparts obtained
using $^{13}$C and $^{15}$N off-resonance $R_{1\rho}$ (Baronti et al., 2020). In addition, similar
exchange parameters were obtained when using variable tilt angles in $R_{1\rho}$
experiments, including tilt angle of 35.3° in which ROE and NOE cross-relaxation
terms cancel (Eichmuller and Skrynnikov, 2005; Weininger et al., 2013; Steiner et
al., 2016). No NOE dips or artifacts were observed in the majority of the $^1$H CEST
or off-resonance $R_{1\rho}$ profiles (Steiner et al., 2016; Dubini et al., 2020; Furukawa et
al., 2021). These results are consistent with a prior off-resonance $^1$H $R_{1\rho}$ studies
showing that even without deuteration, it is feasible to effectively suppress cross-
relaxation between amide and aliphatic protons through selective inversion of
amide protons and use of short spin lock relaxation delays (Lundstrom and Akke,
2005; Schlagnitweit et al., 2018). Nevertheless, NOE effects have been reported
for select sites in $^1$H SELOPE studies of nucleic acids (Schlagnitweit et al., 2018),
and in $^1$H CEST studies of proteins (Bouvignies and Kay, 2012; Sekhar et al., 2016;
Yuwen et al., 2017a; Yuwen et al., 2017b). This underscores the need to carefully
analyze NOE effects, especially for unlabeled samples, in which spin-state-



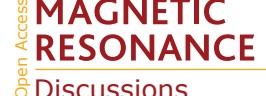

selective magnetization transfer schemes (Yuwen et al., 2017a; Yuwen et al.,
2017b) employing heteronuclei to suppress NOE effects are not feasible.

There are certain conditions in which the Hoogsteen bp becomes the dominant
conformation in duplex DNA.  These include chemically modified bases (Nikolova
et al., 2011), when DNA is in complex with binding partners (Xu et al., 2018), and
for specific sequence contexts under certain experimental conditions (Stelling et
al., 2017).  Based on NMR studies of such duplexes containing Hoogsteen bps,
there should be a sizeable difference ($\Delta\omega$ ~-1 – -2 ppm) between the imino proton
chemical shifts of guanine (G-H1) and thymine (T-H3) in the Hoogsteen versus
Watson-Crick conformation.  These differences should render G-H1 and T-H3
suitable probes of Hoogsteen exchange in unlabeled DNA duplexes provided that
NOE effects can be effectively suppressed.  Imino protons are also attractive
probes given that they are often well-resolved even in 1D [1]H spectra of large RNAs.

Here, we show that high power [1]H CEST SELOPE experiments targeting the imino
protons G-H1 and T-H3 provide facile means for measuring Watson-Crick to
Hoogsteen exchange of G-C and A-T bps in DNA without the need for isotopic
enrichment.  NOE effects are shown to have a negligible contribution as short
(≤100 ms) relaxation delays can be used to characterize the relatively fast ($k_{ex}$ ~



500 to 8,000 s$^{-1}$) Watson-Crick to Hoogsten exchange process (Alvey et al., 2014).
The approach also takes advantage of high-power radio-frequency (RF) fields
recently shown (Rangadurai et al., 2020a) to extend the timescale sensitivity of
CEST to include faster exchange processes that traditionally are more effectively
characterized with the use of $R_{1\rho}$. The high-power $^1$H CEST experiment also
enabled measurement of fast Hoogsteen exchange kinetics ($k_{ex} > 20,000$ s$^{-1}$)
inaccessible to conventional $^{13}$C or $^{15}$N off-resonance $R_{1\rho}$ RD. The $^1$H CEST
experiment opens the door to more comprehensively and systematically exploring
how the Watson-Crick to Hoogsteen exchange process varies with sequence and
structural contexts, and physiological conditions of interest.





## 2 Results

### 2.1 Assessment of NOE effects

We used the SELOPE (Schlagnitweit et al., 2018) experiment (Fig. 1b) to measure

[1]H CEST profiles for G-H1 and T-H3 in unlabeled DNA duplexes (Fig. 2) at 25 °C-

26 °C. We used [1]H CEST rather than $R_{1\rho}$ given the greater ease of collecting

profiles for many spins simultaneously, and given that with the use of high-power

RF fields, CEST can effectively characterize exchange processes over a wide

range of timescales (Rangadurai et al., 2020a). Use of high power RF fields was

recently shown to be important to effectively characterize the comparatively fast

($k_{ex}$ ~ 3,000 s[-1]) Watson-Crick to Hoogsteen exchange process using [13]C and [15]N

CEST experiments (Rangadurai et al., 2020a). Here, we also employed high

power RF fields (> 250 Hz) to optimally characterize Watson-Crick to Hoogsteen

exchange using [1]H CEST.





**Figure 2. DNA and RNA duplexes used in this study.** Also shown are 1D $^{1}$H
spectra of the imino region. The buffer conditions were 25 mM sodium chloride,
15 mM sodium phosphate, 0.1 mM EDTA and 10 % $D_2O$. The pH and temperature
are indicated on each spectrum.


An important consideration when performing $^{1}$H CEST experiments are
contributions due to $^{1}$H-$^{1}$H cross-relaxation, which may give rise to extraneous
NOE dips in the $^{1}$H CEST profiles (Ishima et al., 1998; Lundstrom and Akke, 2005;
Eichmuller and Skrynnikov, 2005; Bouvignies and Kay, 2012; Sekhar et al., 2016;
Yuwen et al., 2017a; Yuwen et al., 2017b). These contributions have been
suppressed in proteins through deuteration (Eichmuller and Skrynnikov, 2005;
Lundstrom and Akke, 2005; Lundstrom et al., 2009; Otten et al., 2010; Hansen et
al., 2012; Weininger et al., 2012), and in $^{15}$N isotopically labelled proteins (Yuwen
et al., 2017a; Yuwen et al., 2017b) and nucleic acids (Wang et al., 2021; Liu et al.,
2020) using spin-state-selective magnetization transfer schemes, and through
selective inversion of protons combined with use of short relaxation times
(Lundstrom and Akke, 2005; Schlagnitweit et al., 2018).

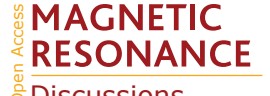

In the SELOPE experiment, imino protons are selectively excited and the
magnetization belonging to non-imino protons is dephased prior to application of
the $B_1$ field.  This helps to suppress cross-relaxation (Yamazaki et al., 1994)
between the imino and non-imino protons (*vide infra*).  In addition, because the
Watson-Crick to Hoogsteen exchange is relatively fast with $k_{ex}$ = ~500 – 8000 s$^{-1}$
at 25 °C (Alvey et al., 2014), we could afford to use a relatively short relaxation
delay of 100 ms which also helped minimize NOE effects (*vide infra*) (Lundstrom
and Akke, 2005; Schlagnitweit et al., 2018).

We initially performed experiments to evaluate contributions from $^1$H-$^1$H cross-
relaxation to the imino $^1$H CEST profiles.  In canonical B-form DNA and A-form
RNA duplexes (Fig. 2), G-H1 is in closest proximity to the partner base C-H4a
(~2.4 Å, Fig. 3a), while T/U-H3 is in closest proximity to the partner A-H2 (~2.8 Å,
Fig. 3a).  Additional proximal protons include imino and H2 protons of neighboring
residues (~3.5-3.6 Å, Fig. 3a).  These short internuclear distances are reflected in
the intensity of cross peaks in 2D [$^1$H, $^1$H] NOESY spectra of nucleic acid duplexes
(Fig. 3b and Fig. S1).  Note that although the amino proton of G-H2a is in proximity
(2.2 Å) to G-H1, while the amino proton of A-H6a is in proximity (2.4 Å) to the
partner T-H3 (Fig. 3a), these amino protons are typically not observable in 1D $^1$H





or 2D [$^1$H,$^1$H] NOESY spectra caused by intermediate exchange due to the
restricted rotation around the C-NH$_2$ bond (Schnieders et al., 2019).

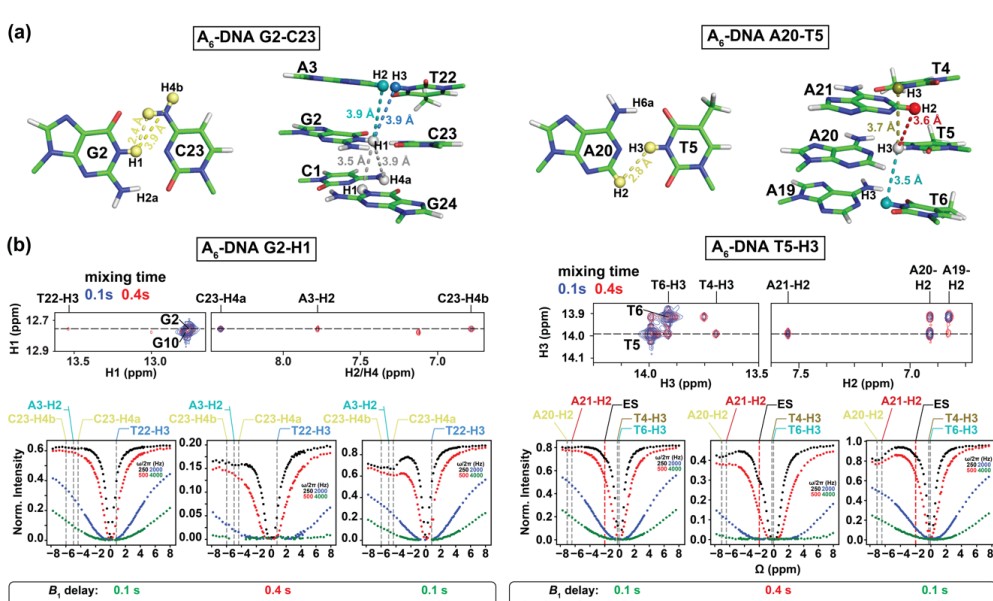

**Figure 3. Analyzing NOE effects in $^1$H CEST profiles.** (a) Distances between
the imino protons of G2-H1 and T5-H3 and nearby protons in the A$_6$-DNA duplex
(PDBID: 5UZF). Note that although the amino proton of G-H2a is in proximity (2.2
Å) to G-H1, while the amino proton of A-H6a is in proximity (2.4 Å) to the partner
T-H3, these amino protons are not observable in 1D $^1$H or 2D [$^1$H,$^1$H] NOESY
spectra caused by intermediate exchange due to the restricted rotation around the
C-NH$_2$ bond (Schnieders et al., 2019). (b) NOE dips in $^1$H CEST profiles for G2-
H1 and T5-H3 in A$_6$-DNA. The NOE diagonal and cross peaks for G2-H and T5-

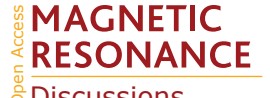

H3 in the 2D [$^1$H, $^1$H] NOESY spectra with mixing time 100 ms (blue) and 400 ms
(red) are shown on the top. The $^1$H CEST profiles for G2-H1 and T5-H3 with
combinations of short (100 ms) and long (400 ms) relaxation delays, with and
without selective excitation (Methods) are shown at the bottom. The ES frequency
(black) obtained from fitting $^1$H CEST profiles with selective excitation and short
relaxation delay (100 ms) as well as frequency positions corresponding to the NOE
cross peaks in the 2D [$^1$H, $^1$H] NOESY spectra (top) are highlighted according to
the color scheme in (a) (bottom). Error bars for CEST profiles in (b), which are
smaller than the data points, were obtained using triplicate experiments, as
described in Methods. RF powers for CEST profiles are color-coded.


$^1$H CEST profiles (Fig. 3b and Fig. S2) for well-resolved imino resonances of $A_6$-
DNA (Fig. 2) were acquired simultaneously in a 1D manner using ~3 hours of
acquisition time on a spectrometer operating at 600 MHz $^1$H frequency equipped
with a cryogenic probe, and using ~1.0 mM unlabeled DNA (Methods). Data were
initially collected at pH = 6.8. Under these near neutral pH conditions, it is
generally not feasible to detect the Watson-Crick to Hoogsteen exchange process
for G-C bps due to the low population of the protonated G-C$^+$ Hoogsteen bp
(Nikolova et al., 2013a). The lack of expected dips for the ES G-C$^+$ Hoogsteen bp





under these conditions provides an opportunity to better assess any extraneous
$^1$H CEST dips arising due to NOE effects. Unlike for G-C bps, the Hoogsteen
exchange should still be detectable for A-T bps under these pH conditions.

Shown in Fig. 3b is a representative imino $^1$H CEST profile measured for G2-H1 in
the well-characterized $A_6$-DNA duplex (Nikolova et al., 2011). Besides the major
dip, no additional dips were visible in the $^1$H CEST profile. The major dip was also
symmetric (Rangadurai et al., 2020a), indicating little to no contribution from
Hoogsteen exchange or NOE effects, as expected for G-C bps under these pH
conditions (Nikolova et al., 2013a). On the other hand, a minor shoulder was
observed in the $^1$H CEST profile of T5-H3 (Fig. 3b). The shoulder occurs at an
offset frequency that does not correspond with any other observable proton
frequency in the $A_6$-DNA duplex and is therefore unlikely to be the result of NOE
effects (Fig. 3a). Rather, as will be described below, the shoulder corresponds to
the ES Hoogsteen bp which is to be expected for the A-T bp at pH = 6.8.

To verify that the dips observed in the $^1$H CEST profile of T5-H3 and other thymine
residues in $A_6$-DNA (see Fig. 4 and S2) do not represent an NOE effect, we
performed $^1$H CEST experiments on a corresponding $A_6$-RNA duplex (Fig. 2).
Unlike in B-form DNA duplexes, G-C$^+$ and A-U Hoogsteen bps are both



308 undetectable in A-form RNA duplexes by off-resonance $^{13}C$ and $^{15}N$ $R_{1\rho}$ RD, most

309 likely due their much lower population ($p_{ES} < 0.04$ %) (Zhou et al., 2016;

310 Rangadurai et al., 2018). If the shoulder observed in the $^{1}H$ CEST profile of T5-

311 H3 in $A_6$-DNA is due to a Hoogsteen ES, and not NOE dips, we would expect to

312 observe a symmetric profile without ES dips for U5-H3 in $A_6$-RNA. Indeed, the

313 corresponding $^{1}H$ CEST profiles for U5-H3 (Fig. 4) and all other uridine and

314 guanine (Fig. S3) imino protons in $A_6$-RNA were symmetric, with no evidence for

315 any asymmetry or shoulder, indicating the absence of exchange and NOE effects.











**Figure 4.** Representative $^1$H CEST profiles measured for A$_2$-DNA (pH 5.4) at 25
°C, A$_5$-DNA (pH 5.2) at 26 °C, A$_6$-DNA (pH 6.8) at 25 °C and A$_6$-RNA (pH 6.8) at
25 °C. Residues with detectable RD, undetectable RD, and overlapped 1D $^1$H
resonances (see Fig. 2) are highlighted in red, blue, and gray circles respectively.
Shown are the fits of the $^1$H CEST data to a 2-state Bloch-McConnell equation with
and without ($k_{ex}$ = Δω = $p_{ES}$ = 0) chemical exchange. Shown below the CEST
profiles are residual (experimental normalized intensity - fitted normalized
intensity) plots. Also shown in inset are the reduced chi-square ($r\chi^2$), and Akaike's
(w*AIC*) and Bayesian information criterion (w*BIC*) weights for fits with exchange
(Methods). The dashed gray lines indicate the Hoogsteen Δω positions in both $^1$H
CEST profiles and in residual plots. Error bars for CEST profiles, which are smaller
than the data points, were obtained using triplicate experiments, as described in
Methods. RF powers for CEST profiles are color-coded.


Therefore, the shoulders in the $^1$H CEST profiles (Fig. 3,4, Fig. S2,3) most likely
rise due to chemical exchange with an ES. This was further confirmed by
evaluating whether fits to the $^1$H CEST profiles show any statistically significant
improvement with the inclusion of exchange, as described below. Based on a
similar analysis, no NOE dips were observable in the $^1$H CEST profiles (Fig. 4,



S2,3) for all other residues in $A_6$-DNA, $A_6$-RNA, and in two other DNA duplexes
across a range of pH and temperature conditions when using selective excitation
and relaxation delay of 100 ms (Fig. 2, Fig. 4, and S2,3).  These results indicate
that any NOE effects between imino and non-imino protons are small under these
experimental conditions.

NOE dips arising from cross-relaxation to neighboring imino protons (Fig. 3a) are
more difficult to assess, as they would be buried within the major dip (Fig. 3b).
However, since no NOE dips were observable for non-imino protons within 2.8 Å
(Fig. 3a), a sizeable cross-relaxation contribution from neighboring imino protons
is unlikely considering they are separated by a longer internuclear distance of ~3.7-
3.9 Å (Fig. 3a), and correspondingly, have weaker intensities in 2D NOESY spectra
(Fig. 3b).  Nevertheless, whether or not these NOE effects are large enough to
impact determination of the exchange parameters was examined (*vida infra*)
through comparison of the exchange parameters derived from fitting the imino $^1$H
CEST profiles with those measured independently using off-resonance $^{13}$C and
$^{15}$N $R_{1\rho}$ RD measurements.

Importantly, upon increasing the relaxation delay to 400 ms, or using a non-
selective $^1$H excitation pulse (pulse **a** in Fig. 1b) with a delay of 100 ms, NOE dips

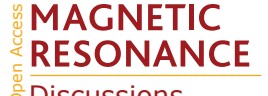

became visible in the $^{1}$H CEST profiles as shown for G2-H1 and T5-H3 (Fig. 3b)
in A$_{6}$-DNA. The dips occurred at the $^{1}$H resonance frequency of nearby protons,
and as expected, were particularly pronounced for the partner C-H4a in the case
of G2-H1 and the partner A-H2 in the case of T5-H3 (Fig. 3b). Nevertheless, the
$^{1}$H CEST profiles acquired with 400 ms delay could be fit when restricting the offset
to the imino proton region (-3 - 3 ppm), and the fitted exchange parameters were
similar to those obtained from fitting profiles with 100 ms relaxation delay in which
no NOE dips were visible (Fig. S4, Table S1). In contrast, the $^{1}$H CEST profiles
measured using non-selective excitation, which had larger NOE dips relative to
using a selective excitation pulse, could not be satisfactorily fit (Fig. S4). These
results underscore the importance of critically evaluating the NOE contributions on
a case-by-case basis (Schlagnitweit et al., 2018) and also suggest that NOE
effects can be effectively suppressed for the canonical duplexes used in this study
provided use of selective excitation and short relaxation delays.

It should be noted that to avoid any complexities due to NOE effects with water
protons or hydrogen exchange, we restricted the offset to -6 ppm – 6 ppm when
analyzing and fitting the $^{1}$H CEST profiles. This is common practice as relatively
narrow offsets (< 4 ppm) were used in prior $^{1}$H CEST studies of both nucleic acids
(Dubini et al., 2020; Wang et al., 2021; Liu et al., 2020) and proteins (Yuwen et al.,



2017a; Yuwen et al., 2017b).  While we did not observe a dip near the water
chemical shift in the $^1$H CEST profile for the internal residue T5-H3, a weak and
broad dip near the water chemical shift was observed in the profile for the near
terminal residue G2-H1 (Fig. S2).  The latter dip could be due to NOEs between
G2-H1 and water protons and/or due to fast hydrogen exchange kinetics.

**2.2    Benchmarking the utility of $^1$H CEST to probe Watson-Crick to**
385            **Hoogsteen exchange in DNA duplexes**

To examine the utility of the SELOPE $^1$H CEST experiment to characterize
Watson-Crick to Hoogsteen exchange, we benchmarked the experiment by
measuring conformational exchange in three DNA duplexes (A$_6$-DNA, A$_2$-DNA and
A$_5$-DNA, Fig. 2) for which we have previously extensively characterized the
Watson-Crick to Hoogsteen exchange using $^{13}$C and $^{15}$N off-resonance $R_{1\rho}$
(Nikolova et al., 2011; Alvey et al., 2014; Shi et al., 2018) and CEST (Rangadurai
et al., 2020a; Rangadurai et al., 2020b) experiments.  We compared the exchange
parameters derived using $^1$H CEST with counterparts derived using $^{13}$C/$^{15}$N $R_{1\rho}$ or
CEST for a variety of G-C and A-T bps across three different DNA duplexes and
varying pH (5.2-6.8) conditions.  All $^1$H CEST experiments were performed using
100 ms relaxation delay and selective excitation.



As expected, for several thymine residues, the imino $^1$H CEST profile was visibly
asymmetric (Fig. 4 and Fig. S2,3), consistent with relatively fast ($k_{ex}$ > 1000 s$^{-1}$)
Watson-Crick to Hoogsteen exchange.  The asymmetry manifests as an upfield
shifted shoulder (e.g. T8-H3 in A$_5$-DNA in Fig. 4) as expected for T-H3 Hoogsteen
chemical shift ($\Delta\omega$ ~-2 ppm) (Nikolova et al., 2011; Xu et al., 2018).  In other cases,
such as T9-H3 in A$_6$-DNA, the asymmetry was less pronounced, and the exchange
contribution was only apparent following comparison of fits with and without
exchange (see Fig. 4).

As expected, at pH = 6.8, the imino $^1$H CEST profiles were symmetric for most
guanine residues consistent with no observable exchange (Fig. 4 and S2,3).
However, the major dip became asymmetric for several guanine residues when
lowering the pH to 5.2 or 5.4, as expected for the Watson-Crick to Hoogsteen
exchange of G-C bps, which is favored at lower pH (Fig. 4 and S3).  All minor dips
occurred at resonance frequencies that did not correspond with any other protons
in the molecule (Fig. 2 and S1,2).  In all cases, the $^1$H CEST profiles could be
satisfactorily fit to a 2-state model with or without exchange, suggesting that any
NOE contribution to the $^1$H CEST profile is likely to be insignificant.

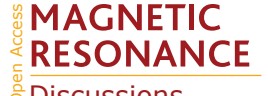

To identify which imino $^1$H CEST profiles have significant chemical exchange
contributions, each profile was subjected to a fit with or without ($\Delta\omega = p_{ES} = k_{ex} =$
0) 2-state chemical exchange (Methods). Akaike information criterion (AIC) and
Bayesian information criterion (BIC) (Burnham and Anderson, 2004) weights were
then used to evaluate whether any improvement in the fit due to inclusion of
chemical exchange was statistically significant (Kimsey et al., 2018; Liu et al.,
2020). The improvement of fit was considered to be statistically significant when
both AIC and BIC weights > 0.995 and the reduced chi-square ($r\chi^2$) is reduced
with the inclusion of exchange. Residual plots were also used to visualize changes
in fit quality (Fig. 4).

Based on the AIC and BIC analysis, all thymine and guanine residues shown
previously to undergo Watson-Crick to Hoogsteen exchange using off-resonance
$^{13}$C and/or $^{15}$N $R_{1\rho}$ under these experimental conditions, also showed statistically
significant improvements when fitting the $^1$H CEST profiles with the inclusion of
chemical exchange (Fig. 4 and S2,3). On the other hand, all guanine residues
including G2 and G11 in $A_6$-DNA and G11 in $A_2$-DNA, which did not show signs of
Hoogsteen exchange in off-resonance $^{13}$C and/or $^{15}$N $R_{1\rho}$ (Nikolova et al., 2011;
Shi et al., 2018) under these experimental conditions also did not show statistically



significant improvements when fitting their $^1$H CEST profiles with the inclusion of
chemical exchange (Fig. 4 and S2,3).

Interestingly, a few residues including T5, T6, T7 and T22 in A$_6$-DNA, T18, G6 and
G20 in A$_2$-DNA (Fig. S2,3), showed exchange based on $^1$H CEST but did not show
evidence for Hoogsteen exchange based on prior off-resonance $^{13}$C and/or $^{15}$N $R_{1\rho}$
experiments (Nikolova et al., 2011; Alvey et al., 2014; Shi et al., 2018).  As will be
elaborated in the following section, these data provide new insights into the
Watson-Crick to Hoogsteen exchange process, and suggest that at least in some
cases, $^1$H CEST can exceed the detection limits of $^{13}$C/$^{15}$N based methods.

In addition, T18 and G20 in A$_2$-DNA were difficult to probe using $^{13}$C RD due to
spectra overlap (Nikolova et al., 2011) but could easily be measured using $^1$H
CEST (Fig. 2, 4 and S3).  In contrast, other residues such as T8 and T4 in A$_6$-DNA,
T4 and T22 in A$_2$-DNA, and G10 and G11 in A$_5$-DNA could be targeted for $^{13}$C or
$^{15}$N RD measurements (Nikolova et al., 2011; Alvey et al., 2014) but could not be
measured by $^1$H CEST due to overlap in the 1D $^1$H imino spectra (Fig. 2).  This
highlights the complementarity of $^1$H and $^{13}$C/$^{15}$N RD in characterizing Watson-
Crick to Hoogsteen exchange.

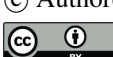



To assess how well the exchange parameters are determined by the $^1$H CEST
data, we subjected the $^1$H CEST profiles for residues T7 ($k_\text{ex}/\Delta\omega \sim 0.2$), T9 ($k_\text{ex}/\Delta\omega$
$\sim 0.82$) and T22 ($k_\text{ex}/\Delta\omega \sim 3.5$) which exhibit exchange on the slow, intermediate,
and fast timescale (Rangadurai et al., 2019b) respectively, to a degeneracy
analysis. We computed the reduced chi-square ($r\chi^2$) for a 2-state fit as a function
of varying $k_\text{ex}$, $\Delta\omega$ or $p_\text{ES}$. In all cases, the $r\chi^2$ values increased significantly (up
to 10-fold) when varying $k_\text{ex}$, $\Delta\omega$ or $p_\text{ES}$ by 3-fold (Fig. S5), indicating that the
exchange parameters are well-defined by the $^1$H CEST data.

To verify that the exchange process sensed by $^1$H CEST does indeed correspond
to Watson-Crick to Hoogsteen exchange, we compared the exchange parameters,
$p_\text{ES}$ and $k_\text{ex}$, derived from a 2-state fit of the data to values determined previously
using off-resonance $^{13}$C and/or $^{15}$N $R_{1\rho}$ (Nikolova et al., 2011; Shi et al., 2018;
Alvey et al., 2014) for Hoogsteen dynamics (Fig. 5a and Table S1). In total, we
were able to compare 13 data points from $^1$H CEST and $^{13}$C/$^{15}$N $R_{1\rho}$ for three
different duplexes under different conditions of temperature and pH (Fig. 2,5a).





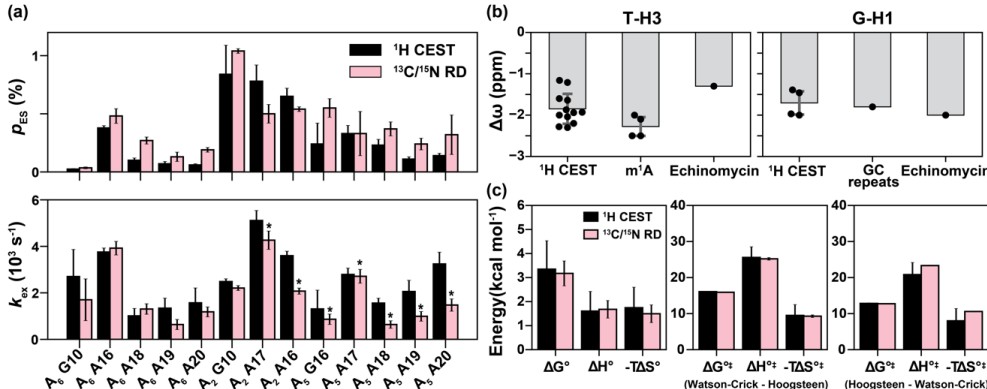


**Figure 5.** Comparison of exchange parameters for the Watson-Crick to

Hoogsteen exchange obtained from $^1$H CEST and $^{13}$C/$^{15}$N $R_{1\rho}$. (a) Comparison of

exchange parameters ($k_{ex}$ and $p_{ES}$) measured using $^1$H CEST with counterparts

previously reported using $^{13}$C/$^{15}$N off-resonance $R_{1\rho}$ (Nikolova et al., 2011; Alvey

et al., 2014; Shi et al., 2018). $^{13}$C RD data for A18, A19 and A20 were measured

using off-resonance $R_{1\rho}$ in this study (Fig. S7). Small systematic deviations in $k_{ex}$

for the values indicated with asterisks could be due to small differences in

temperature (< 0.8ºC) across different spectrometers. Bps are specified by the

corresponding purine residue. (b) Comparison of the Δω obtained from fitting $^1$H

CEST profiles for T-H3 and G-H1 (Table S1) with the values expected for a

Watson-Crick to Hoogsteen transition based on duplexes in which A-T or G-C$^+$

Hoogsteen bps were rendered the dominant state, by using $N^1$-methylated adenine

(m$^1$A) (Nikolova et al., 2011; Sathyamoorthy et al., 2017; Rangadurai et al.,

2020b), by binding of the  drug (echinomycin) to a DNA duplex (Xu et al., 2018),



or through use of GC repeat sequences (GC repeats) that predominantly form
Hoogsteen bps at low pH (Stelling et al., 2017).  (c) Comparison of free energy
(ΔG°), enthalpy (ΔH°) and entropy (-TΔS°, T = 25 °C) of the Watson-Crick to
Hoogsteen transition, and the activation free energy (ΔG°‡), enthalpy (ΔH°‡) and
entropy (-TΔS°‡, T = 25 °C) for Watson-Crick to Hoogsteen (Watson-Crick -
Hoogsteen) and Hoogsteen to Watson-Crick (Hoogsteen - Watson-Crick)
transitions measured using $^1$H CEST in this study and using $^{13}$C $R_{1\rho}$ from Nikolova
*et al* (Nikolova et al., 2011).  The energetics in (c) were measured for the Watson-
Crick to Hoogsteen transition of A16-T9 in $A_6$-DNA at pH 6.8.  Errors in (a) were
fitting errors of $^1$H CEST, calculated as described in Methods or errors of $^{13}$C/$^{15}$N
$R_{1\rho}$ calculated using a Monte-Carlo scheme as described previously (Rangadurai
et al., 2019b).  Errors in (b) are the standard deviations of data points (shown as
black dots) in each category.  Error bars in (c) were propagated from the errors in
the exchange parameters obtained from $^1$H CEST or $^{13}$C/$^{15}$N $R_{1\rho}$.


Indeed, the $p_{ES}$ and $k_{ex}$ values derived using $^1$H CEST were in very good
agreement with their off-resonance $^{13}$C and/or $^{15}$N $R_{1\rho}$ counterparts (Fig. 5a).  The
differences between $k_{ex}$ and $p_{ES}$ measured using the two methods was often within
error with the largest differences being <3-fold.  A small and systematic difference



in $k_{ex}$ was observed for a subset of the data (Fig. 5a), and this might be due to
small temperature differences (<0.8ºC) between spectrometers.  Importantly, the
ES imino $^1$H chemical shifts deduced from a 2-state fit of the $^1$H CEST profiles
($\Delta\omega_{A-T}$ = ~-1 to -2 ppm and $\Delta\omega_{G-C}$ = ~-1.5 to -2.0 ppm) were also in good agreement
with the expected range of values ($\Delta\omega$ = -1 to -2 ppm) for Hoogsteen bps (Fig. 5b)
based on studies of duplexes containing Hoogsteen bps as the dominant
conformation (Nikolova et al., 2011; Stelling et al., 2017; Xu et al., 2018;
Rangadurai et al., 2020b).

As an additional test, we also measured temperature-dependent (5 °C, 10 °C,
20 °C, 25 °C, 30 °C and 45 °C) $^1$H CEST profiles for $A_6$-DNA at pH 6.8 (Fig. S2),
and then used the temperature dependence of the fitted kinetic rate constants ($k_1$
and $k_{-1}$) to determine the standard and activation enthalpy and entropy changes
for the Watson-Crick to Hoogsteen transition (Fig. S6).  These values were in
excellent agreement with those measured from off-resonance $^{13}$C $R_{1\rho}$ (Nikolova et
al., 2011) (Fig. 5c), further supporting the robustness of the $^1$H CEST methodology.

**2.3   New insights into Hoogsteen breathing**



$^1$H CEST profiles for some residues show detectable exchange contributions when
corresponding $^{13}$C/$^{15}$N RD measurements do not or show only weak exchange.
This suggests that $^1$H CEST can provide additional insights into Watson-Crick to
Hoogsteen exchange and extend the detection limits of conventional $^{13}$C/$^{15}$N RD
measurements.

For example, using $^1$H CEST it was feasible to measure Watson-Crick to
Hoogsteen exchange for T5-H3, T6-H3, and T7-H3 (Fig. S2) within the middle of
the A-tract motif (defined as $A_n$-tract with n>3) in $A_6$-DNA.  These residues had
previously exhibited only weak on-resonance $^{13}$C $R_{1\rho}$ RD, and as a result, no off-
resonance $R_{1\rho}$ data were ever recorded (Nikolova et al., 2011).  Based on the $^1$H
CEST measurements, residues within the A-tract motif have ten-fold lower
Hoogsteen population ($p_{ES}$ = 0.06±0.01 %-0.09±0.03 %) relative to other A-T bps
in $A_6$-DNA ($p_{ES}$ > ~0.10 %) (Table S1).  These represent the lowest A-T Hoogsteen
populations ever recorded to date in duplex DNA (Table S1).  The exchange
kinetics were also 2-fold slower ($k_{ex}$ ~1000 s$^{-1}$) for the A-tract residues relative to
other A-T bps ($k_{ex}$ > 2000 s$^{-1}$) in $A_6$-DNA (Table S1).  Interestingly, the suppression
of Hoogsteen dynamics within the A-tract motif appears to be A-tract length
dependent, with both the Hoogteen population and exchange kinetics increasing
slightly for similar bps in $A_5$-DNA (Table S1).  The suppression of Hoogsteen





dynamics within A-tracts is consistent with prior studies showing them to be more
rigid and stiff motifs relative to scrambled DNA (Nikolova et al., 2012b).    We
verified these $^1$H CEST derived exchange parameters for A-tract residues in A$_6$-
DNA by performing off-resonance $^{13}$C $R_{1\rho}$ measurements (Fig. S7) on uniformly
$^{13}$C/$^{15}$N labeled A$_6$-DNA and did indeed observe the expected RD with $p_{ES}$ and $k_{ex}$
values similar (difference <3-fold, Fig. 5a) to those measured using $^1$H CEST.
These prospective tests of the $^1$H CEST data using off-resonance $^{13}$C/$^{15}$N $R_{1\rho}$ RD
data further support the methodology.

The ability to characterize fast exchange kinetics has long been a motivation for
using $^1$H in RD experiments to characterize conformational exchange (Ishima et
al., 1998; Ishima and Torchia, 2003; Eichmuller and Skrynnikov, 2005; Lundstrom
and Akke, 2005; Otten et al., 2010; Hansen et al., 2012; Smith et al., 2015; Steiner
et al., 2016; Furukawa et al., 2021).  Indeed, $^1$H CEST made it possible to measure
fast Watson-Crick to Hoogsteen exchange kinetics which were undetectable by
off-resonance $^{13}$C $R_{1\rho}$.  In particular, it was possible to measure Watson-Crick to
Hoogsteen exchange for T22 in A$_6$-DNA with $k_{ex} > 20,000$ s$^{-1}$ (Fig. S2 and Table
S1), which is the fastest ever recorded Hoogsteen exchange process at 25 °C
(Table S1).  In contrast, the off-resonance $^{13}$C $R_{1\rho}$ RD profiles reported for this
residue in prior studies were flat (Nikolova et al., 2011; Shi et al., 2018), and



simulations show that such an exchange process is too fast for reliable detection
using $^{13}$C $R_{1\rho}$ (Fig. S8a).  Similarly, it was feasible to measure Watson-Crick to
Hoogsteen exchange for G6 ($p_{ES}$ ~0.3 %, $k_{ex}$ ~3000 s$^{-1}$) in A$_2$-DNA using $^1$H CEST
yet no off-resonance $^{13}$C $R_{1\rho}$ RD on C1' was previously detected (Shi et al., 2018),
which based on simulations, was likely due to a combination of exchange kinetics
and small $\Delta\omega$ value (Fig. S8b).

One of the potential utilities of the $^1$H CEST experiment is the measurement of very
fast exchange kinetics at high temperatures and in a manner insensitive to melting
of duplexes, shown previously to complicate analysis of Hoogsteen exchange
using $^{13}$C and $^{15}$N RD (Shi et al., 2019).  Melting of duplexes should not yield any
exchange dips around the imino $^1$H region given that the imino protons of single-
stranded species (ssDNA) exchange rapidly with solvent.

We therefore measured $^1$H CEST profiles for A$_6$-DNA at 45 °C (Fig. S2), in which
the ssDNA population is ~10 % (Shi et al., 2019).  We did not observe any evidence
for the ssDNA species in the $^1$H CEST profiles.  Instead, we were able to observe
ultra-fast ($k_{ex}$ ~ 10,000 s$^{-1}$, see Table S1) Hoogsteen exchange which could not
previously be detected by $^{13}$C or $^{15}$N RD experiments at the same temperature (Shi
et al., 2019).

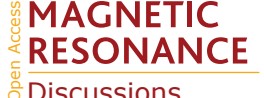


Taken together, these results demonstrate that the [1]H CEST experiment broadens
the range of populations and exchange rates over which Hoogsteen breathing can
be effectively characterized.
**3    Discussion**
Building on prior studies showing the utility of the SELOPE [1]H RD experiment in
measuring conformational exchange in unlabeled RNA (Schlagnitweit et al., 2018)
and DNA (Furukawa et al., 2021; Dubini et al., 2020), our study establishes the
utility of high-power [1]H CEST SELOPE as a facile means for measuring the
Watson-Crick to Hoogsteen exchange process in nucleic acids without the need
for isotopic enrichment. The methodology is supported by the very good
agreement observed between the measured exchange parameters and values
measured independently using $^{13}$C and/or $^{15}$N $R_{1\rho}$ for a variety of bps in three
duplexes under different conditions of temperature and pH, as well as by the good
agreement seen between the imino [1]H chemical shifts and those expected based
on duplexes containing Hoogsteen bps as the dominant GS conformation. The
high throughput nature of the experiment and simple sample requirements enabled
us to measure Hoogsteen dynamics for 37 data points corresponding to 22 distinct



bps for three different pH conditions and seven different temperatures (Table S1),
the largest collection of Hoogsteen dynamics from a single study to date. We
envision using the $^1$H CEST SELOPE experiments to pre-screen DNA duplexes
and to perform follow-up $^{13}$C and $^{15}$N RD experiments to confirm any interesting
outliers, particularly regions showing substantially elevated Hoogsteen dynamics.

An important consideration when applying $^1$H CEST to the study of chemical
exchange are contributions due to $^1$H-$^1$H cross-relaxation originating from cross
relaxation, which may give rise to extraneous NOE dips that complicate data
analysis (Yuwen et al., 2017a; Bouvignies and Kay, 2012; Eichmuller and
Skrynnikov, 2005). These contributions have been shown to be significant in
proteins particularly when characterizing slow exchange ($k_{ex} < 200$ s$^{-1}$)
necessitating use of relatively long relaxation delays (Bouvignies and Kay, 2012).
Consistent with prior studies of nucleic acids (Schlagnitweit et al., 2018; Steiner et
al., 2016; Baronti et al., 2020) and proteins (Lundstrom and Akke, 2005), our
results indicate that NOE effects involving imino protons can be effectively
suppressed for DNA and RNA duplexes in the $^1$H CEST experiments through
selective excitation provided that the relaxation delays are short on the order of
100 ms (Fig. 3b). However, because NOE dips were clearly visible when using
400 ms relaxation delay, care should be exercised on a case-by-case basis to

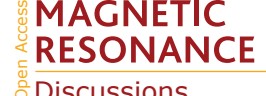

evaluate NOE effects (Fig. 3b), which may also be more substantial for certain
non-canonical motifs.  In addition to cross-referencing the dip with 2D [$^1$H, $^1$H]
NOESY spectra, testing whether the dip increases in magnitude without selective
excitation can help to distinguish between dips due to an ES versus NOE effects.

Prior studies showed that Watson-Crick to Hoogsteen bp transitions exhibit large
variations in the forward rate constants ($k_1$) while the backward rate constants ($k_{-1}$
) is relatively constant across different sequence contexts, consistent with a late
transitional state (Alvey et al., 2014).  We observe a similar trend in which $k_{-1}$ varied
<5-fold while $k_1$ varied by ~50-fold (Fig. S9).  The $^1$H CEST data also revealed
significantly lower Hoogsteen abundance ($p_{ES}$ < 0.1 %) in addition to slower
exchange kinetics ($k_{ex}$ ~1,000 s$^{-1}$) within A-tract motifs (Nikolova et al., 2011; Alvey
et al., 2014), while also reinforcing prior data (Xu et al., 2018) suggesting increased
exchange kinetics near terminal ends.  Collectively, these data show that the
Hoogsteen population can vary by as much as ~14-fold while $k_{ex}$ can vary by ~20-
fold only due to changes in sequence and positional context (Table S1).  These
strong sequence and position dependencies could play important roles in
biochemical processes acting on DNA.



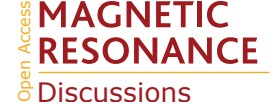

A recent study (Furukawa et al., 2021) reported on-resonance imino $^1$H $R_{1\rho}$ RD for
a guanine residue in a DNA duplex at pH = 7.5, T= 30 °C, and in 150 mM NaCl.
Because off-resonance measurements were not performed, only $k_{ex} \sim$ 10,000 s$^{-1}$
could be determined while the values of Δω and $p_{ES}$ were not determined. The
study noted that a Hoogsteen bp as the ES was unlikely given that G-C$^+$ Hoogsteen
bps are disfavored at pH= 7.5 and because the observed rate of exchange ($k_{ex} \sim$
10,000 s$^{-1}$) was much faster than is typically observed for Watson-Crick to
Hoogsteen exchange. Instead, the data were interpreted as evidence for a base
opened state. However, the observed rate of exchange $k_{ex}$ ~10,000 s$^{-1}$ falls
comfortably within the range of values measured here for Watson-Crick to
Hoogsteen exchange using $^1$H CEST at similar pH conditions. For example, for
the G10-C15 bp in A$_6$-DNA at the same temperature and pH = 6.8, $k_{ex}$ for Watson-
Crick to Hoogsteen exchange was ~6,000 s$^{-1}$ (Fig. 4 and Table S1). Similar
Watson-Crick to Hoogsteen exchange parameters ($p_{ES}$ ~0.05 % and $k_{ex}$ ~2000 s$^{-}$
$^1$) were recently reported for this bp at 25 °C and pH 6.8 using cytosine amino $^{15}$N
RD (Rangadurai et al., 2019a) and the ES $\Delta\omega_{C-N4}$ = -9 ppm was shown to be in
excellent agreement with values expected for a G-C$^+$ Hoogsteen bp. In addition,
based on hydrogen exchange measurements, $p_{ES}$ ~0.00001 % to 0.01 % and $k_{ex}$
($k_{cl}$ + $k_{op}$, $k_{cl}$ and $k_{op}$ are the base closing and opening rate constant, respectively)
~ 10$^5$ to 10$^7$ s$^{-1}$ for the base-opened ES, and this process should fall outside RD





detection (Gueron and Leroy, 1995; Gueron et al., 1987; Leroy et al., 1988; Leijon
and Graslund, 1992; Snoussi and Leroy, 2001). Therefore, the ES detected by
Furukawa *et al* (Furukawa et al., 2021) is more likely a Hoogsteen bp.

In conclusion, by obviating the need for isotopic enrichment, the $^1$H CEST
experiment expands the scope of characterizing Watson-Crick to Hoogsteen
exchange in nucleic acids by NMR. We are presently applying the experiment to
map the sequence dependence of Hoogsteen breathing dynamics and
systematically, how it varies with pH, salt, and crowding, and following the
introduction of lesions, mismatches, and molecules that bind to the DNA.



**4    Methods**
**4.1    Sample preparation**
*Unlabeled DNA and RNA oligonucleotides*: Unmodified DNA oligonucleotides
were purchased from Integrated DNA Technologies with standard desalting
purification. RNA oligonucleotides were synthesized using a MerMade 6 Oligo
Synthesizer employing 2'-tBDSilyl protected phosphoramidites (n-acetyl protected
rC, rA and rG, and rU phosphoramidites were purchased from Chemgenes) and 1
µmol standard synthesis columns (1000 Å) (BioAutomation). RNA
oligonucleotides were synthesized with the final 5'-protecting group, 4,4'-
dimethoxytrityl (DMT) retained. RNA oligonucleotides were cleaved from columns
using 1 ml AMA (1:1 ratio of 30 % ammonium hydroxide and 30 % methylamine)
and incubated at room temperature for 2 hours. The sample was then air-dried
and dissolved in 115 µL DMSO, 60 µL TEA, and 75uL TEA.3HF, and then
incubated at T = 65 °C for 2.5 hours to remove 2'-O protecting groups. The Glen-
Pak RNA cartridges (Glen Research Corporation) were then used to purify the
samples followed by ethanol precipitation.

*Labeled DNA oligonucleotides*: The uniformly $^{13}$C, $^{15}$N labeled $A_6$-DNA sample was
prepared using chemically synthesized DNA (purchased from IDT), Klenow

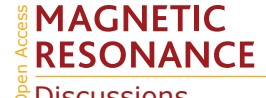

fragment DNA polymerase (New England Biolab) and $^{13}C/^{15}N$ isotopically labeled
dNTPs (Silantes) using the Zimmer and Crothers method (Zimmer and Crothers,
1995). The oligonucleotide was purified using 20 % 29:1 polyacrylamide
denaturing gel with 8 M urea, 20 mM Tris borate and 1 mM EDTA, and then using
electro-elution (Whatmann, GE Healthcare) in 40 mM Tris Acetate and 1 mM
EDTA, followed by ethanol precipitation.

*Sample annealing and buffer exchange*: DNA/RNA oligonucleotides were re-
suspended in water (200-500 µM). To prepare duplex samples, equimolar
amounts of the constituent single stranded DNA/RNA samples were mixed and
then heated at T = 95 °C for ~5 min followed by cooling at room temperature for
~1 hour. All samples were exchanged three times into the desired buffer using
centrifugal concentrators (4 mL, Millipore Sigma). 10 % $D_2O$ (Millipore Sigma) was
added to the samples prior to the NMR measurements.

*Sample concentrations and buffer conditions*: Unless mentioned otherwise, the
NMR buffer contains 25 mM sodium chloride, 15 mM sodium phosphate, 0.1 mM
EDTA and 10 % $D_2O$. Sample concentrations and buffer pH: $A_6$-DNA, 1.0 mM, pH
6.8; $A_2$-DNA, 1.0 mM, pH 5.4; $A_5$-DNA, 0.2 mM, pH 5.2; $A_6$-RNA, 0.5 mM, pH 6.8.
Concentration was estimated by measuring the absorbance of the sample at

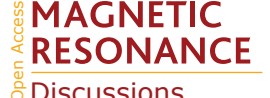

260nm and using extinction coefficients from the ADT Biol Oligo calculator
(https://www.atdbio.com/tools/oligo-calculator).

**4.2   NMR spectroscopy**
All NMR experiments were performed on a 600 BrukerAvance 3 spectrometer
equipped with a triple-resonance HCN cryo-genic probe.  The NMR data were
processed and analyzed with NMRpipe (Delaglio et al., 1995) and SPARKY (T.D.
Goddard and D.G. Kneller, SPARKY 3, University of California, San Francisco).

*Resonance assignments*: Imino resonances were assigned using a combination
of 2D [$^1$H, $^1$H] NOESY and [$^{15}$N,$^1$H] SOFAST-HMQC (Sathyamoorthy et al., 2014)
experiments.   Assignments for $A_6$-DNA, $A_2$-DNA and $A_6$-RNA were reported
previously (Sathyamoorthy et al., 2017; Zhou et al., 2016; Nikolova et al., 2011).
The [$^1$H, $^1$H] NOESY spectrum for $A_5$-DNA is shown in Fig. S1.

*$^1$H CEST*: The pulse sequence was shown in Fig. 1b, and was adapted from
Schlagnitweit *et al* (Schlagnitweit et al., 2018).  Relaxation delays $T_{EX}$ = 100 ms
was used for all $^1$H CEST measurements at low temperatures (5 °C – 30 °C), while
a shorter $T_{EX}$ = 80 ms was used for high (45 °C) temperature measurements.  A

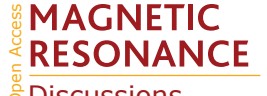

longer $T_{EX}$ = 400 ms was used to illustrate artefacts arising due to NOE dips (Fig.
3b). RF power and offset combinations used in the CEST measurements are given
in Table S2. Calibration of RF field powers for the [1]H CEST measurements was
performed as described previously (Rangadurai et al., 2019b) using the same
pulse sequence. Field inhomogeneity was also measured (Fig. S10) using the
same sequence and the procedure as described previously (Guenneugues et al.,
1999). [1]H inhomogeneity was measured by performing on-resonance [1]H CEST
experiments on G2-H1 of $A_6$-DNA, chosen as it does not experience
conformational exchange. The longest relaxation delay used for the
measurements were 10 s, 2 s, 1 s, 0.4 s, 0.1 s and 0.04 s for RF fields 10 Hz, 50
Hz, 100 Hz, 200 Hz, 1000 Hz and 4000 Hz, respectively. The resulting nutation
curve was Fourier transformed and was fit to a gaussian function (blue lines in Fig.
S10) to extract the full-width at half-maximum, which was used for defining the
inhomogeneity as described previously (Guenneugues et al., 1999). The selective
pulse was set to be off (Fig. 3b) by replacing pulse **a** (Fig. 1b) with a non-selective
[1]H hard 90° pulse.

*Fitting of [1]H CEST data*: When performing 2-state CEST fitting with and without
exchange, we restricted the offset to -6 to 6 ppm for [1]H CEST experiment with
relaxation delay ≤ 100 ms, and to -3 to 3 ppm for experiments with relaxation delay



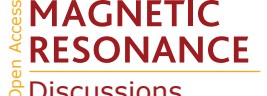

= 400 ms, to obviate any potential effects from $^1$H-$^1$H cross-relaxation artifacts (Fig.
3b). Peak intensities of all imino protons in the 1D spectra as a function of RF
power and offset frequency were extracted using NMRPipe (Delaglio et al., 1995).
The peak intensity at a given RF power and offset is normalized by the average
peak intensity over the triplicate CEST measurements with zero relaxation delay
under the same RF power. The uncertainty in the measured peak intensity at each
offset frequency and RF power combination was assumed to be equal to the
standard deviation of the peak intensities for triplicate CEST experiments with zero
relaxation delay under the same RF power (Zhao et al., 2014; Shi et al., 2019).
CEST profiles were generated by plotting the normalized intensity as a function of
offset $\Omega = \omega_{RF} - \omega_{obs}$ where $\omega_{obs}$ is the Larmor frequency of the observed
resonance and $\omega_{RF}$ is the angular frequency of the applied RF field. RF field
inhomogeneity (Fig. S10) was taken into account during CEST fitting as described
previously (Rangadurai et al., 2020a). The normalized CEST profiles were then fit
via numerical integration of the Bloch-McConnell (B-M) equations as described
previously (Rangadurai et al., 2020a). Fitting of CEST profiles without exchange
(Fig. 4, Fig. S2-4) was performed by setting $p_{ES} = k_{ex} = \Delta\omega = 0$. Errors in exchange
parameters were set to be equal to the fitting errors which were obtained as the
square root of the diagonal elements of the covariance matrix. Reduced chi-



square ($r\chi^2$) was calculated to assess the goodness of fitting (Rangadurai et al.,
2019b).  The residual sum of squares (RSS) was computed as follows
$$RSS = \sum_{i=1}^{n}\left(I_i^{fit} - I_i^{exp}\right)^2 \tag{1}$$
where $I_i^{fit}$ and $I_i^{exp}$ are the $i$th fit and experimentally measured intensity in the
CEST profile respectively, and the summation is over all RF power and offset
combinations (N).

Model selection for fits with and without exchange (Fig. 4, Fig. S2-4) was
performed by computing AIC and BIC weights as follows (Burnham and Anderson,

779    2004):


$$AIC = \begin{cases} Nln\left(\dfrac{RSS}{N}\right) + 2K, & when \dfrac{N}{K} \geq 40 \\ Nln\left(\dfrac{RSS}{N}\right) + 2K + \dfrac{2K(K+1)}{N-K-1}, & when \dfrac{N}{K} < 40 \end{cases} \tag{2}$$

$$wAIC = \frac{e^{-0.5\Delta AIC}}{1 + e^{-0.5\Delta AIC}} \tag{3}$$

$$BIC = Nln\left(\frac{RSS}{N}\right) + Kln(N) \tag{4}$$

$$wBIC = \frac{e^{-0.5\Delta BIC}}{1 + e^{-0.5\Delta BIC}} \tag{5}$$






Where *K* is the number of floating parameters when fitting and $\Delta AIC/\Delta BIC$ are the
differences between two AIC values (fitting without and with exchange). The AIC
(*wAIC*+ex) and BIC (*wBIC*+ex) weights for fits with exchange are reported in Fig. 4
and Fig. S2-4. The improvement in the fit was considered statistically significant if
both *wAIC*+ex and *wBIC*+ex values are > 0.995, and $r\chi^2$ is reduced with the inclusion
of exchange. For some resonances, the improvement in the fit with exchange are
statistically significant but the resulting exchange parameters are not reliable and
have large errors (see Fig. S2,3). For T4 in $A_5$-DNA, $p_{ES}$ = 0.2±0.1 % measured
using $^1$H CEST was ~10-fold smaller than $p_{ES}$ = 2.7±1.5 % measured previously
using $^{15}$N RD (Alvey et al., 2014), whereas $k_{ex}$ (~3000 s$^{-1}$) was is in good
agreement. However, simulations show that due to the small Δω for $^{15}$N (~1 ppm)
and fast exchange kinetics $k_{ex}$ (~3000 s$^{-1}$) the $p_{ES}$ and Δω are not well-determined
by the $^{15}$N RD data (Fig. S6c). For this reason, this data point was excluded for $^1$H
CEST and $^{13}$C/$^{15}$N RD comparison (Fig. 5a).

*Off-resonance $^{13}$C $R_{1\rho}$ relaxation dispersion*: $^{13}$C $R_{1\rho}$ experiments were performed
using 1D $R_{1\rho}$ schemes as described previously (Nikolova et al., 2012a; Nikolova
et al., 2011; Hansen et al., 2009). The spin-lock powers and offsets are listed in
Table S3. The spin-lock was applied for a maximal duration < 60 ms to achieve



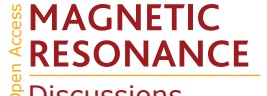

~70 % loss of peak intensity at the end of relaxation delay.  Off-resonance $R_{1\rho}$
profiles (Fig. S8) were generated by plotting $(R_2 + R_{ex}) = (R_{1\rho} - R_1\cos^2\theta)/\sin^2\theta$,
where $\theta$ is the angle between the effective field of the observed resonance and the
z-axis, as a function of $\Omega_{eff}/2\pi$, where $\Omega_{eff} = \omega_{obs} - \omega_{RF}$, where $\omega_{obs}$ is the Larmor
frequency of the spin and $\omega_{RF}$ is the carrier frequency of the applied spin-lock.

*Fitting of $^{13}C$ $R_{1\rho}$ data:* 1D peak intensities were measured using NMRpipe
(Delaglio et al., 1995).  $R_{1\rho}$ values for a given spin-lock power and offset were
calculated by fitting the intensities as a function of delay time to a mono-
exponential decay (Kimsey et al., 2015).  A Monte-Carlo approach was used to
calculate the uncertainties of $R_{1\rho}$ (Bothe et al., 2014).  Alignment of initial
magnetization during the Bloch-McConnell fitting was performed based on the
$k_{ex}/\Delta\omega$ value (Rangadurai et al., 2019b).  Chemical exchange parameters were
obtained by fitting experimental $R_{1\rho}$ values to numerical solutions of a 2-state
Bloch-McConnell (B-M) equations (Mcconnell, 1958).  A Monte-Carlo approach
was used to calculate the errors of exchange parameters (Bothe et al., 2014) .
Reduced chi-square ($r\chi^2$) was calculated to assess the goodness of fitting
(Rangadurai et al., 2019b).

**4.3   Thermodynamic Analysis**

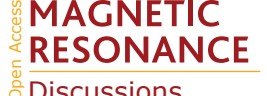

The observed temperature dependence of $k_1$, $k_{-1}$ for the Watson-Crick to
Hoogsteen exchange measuring using $^1$H CEST were fit to a modified van't Hoff
equation that accounts for statistical compensation effects and assumes a smooth
energy surface as described previously (Nikolova et al., 2011; Coman and Russu,

832 2005):


$$\ln\left(\frac{k_i(T)}{T}\right) = \ln\left(\frac{k_B\kappa}{h}\right) - \frac{\Delta G_i^{\circ\,T}(T_{hm})}{RT_{hm}} - \frac{\Delta H_i^{\circ\,T}}{R}\left(\frac{1}{T} - \frac{1}{T_{hm}}\right) \tag{6}$$


$k_i$ (i = 1, -1) is the forward and backward rate constants, $\Delta G_i^{\circ\,T}(T)$ and $\Delta H_i^{\circ\,T}$ are the
free energy (at temperature T, in Kelvin) and enthalpy of activation (i = 1) or
deactivation (i = -1) respectively. $R$ is the universal gas constant (kcal mol$^{-1}$ K$^{-1}$)
and $T_{hm}$ is the harmonic mean of the experimental temperatures ($T_i$ in K) computed
as $T_{hm} = n/\sum_{i=1}^{n}(1/T_i)$ , $k_B$ is the Boltzmann's constant (J K$^{-1}$), $\kappa$ is the
transmission coefficient (assumed to be unity) and $h$ is the Planck constant (J s).

The goodness-of-fit indicator $R^2$ (coefficient of determination) (Fig. S6) between
the measured and fitted rate constants was calculated as follows: $R^2 = 1 -$
$\frac{SS_{res}}{SS_{total}}, SS_{res} = \sum\left(k_{i,fit} - k_{i,exp}\right)^2, SS_{total} = \sum\left(k_{i,exp} - \overline{k_{i,exp}}\right)^2.$ $k_{i,fit}$ and $k_{i,exp}$ (i
= 1, -1) are fitted and experimentally measured rate constants. $\overline{k_{i,exp}}$ is the mean

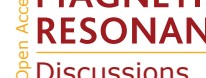

of all $k_{i,exp}$. Errors of fitting for $\Delta G_i^{\circ\,T}$ and $\Delta H_i^T$ were calculated as the square root
of the diagonal elements of the covariance matrix. $T\Delta S_i^T$ is calculated as $\Delta H_i^T -$
$\Delta G_i^{\circ\,T}$.



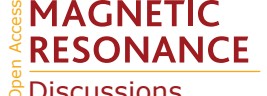

**Data and code availability.** The data that support this study are contained in the
published article (and its Supplementary Information) or are available from the
corresponding author on reasonable request. The python scripts for $^1$H CEST data
fitting are available at https://github.com/alhashimilab/1H-CEST.

**Author contributions.** BL, AR, and HMA conceived the project and experimental
design. BL, AR, and HS prepared the samples and set up the imino $^1$H CEST
experiment. BL performed $^1$H CEST experiments and data analysis. HS
performed $^{13}$C $R_{1\rho}$ experiments. HMA, BL, and AR wrote the manuscript with
critical input from HS.

**Competing interests.** The authors declare that they have no conflict of interest.

**Acknowledgments.** We thank Prof. Katja Petzold for sharing the $^1$H CEST pulse
sequence. We thank Dr. Or Szekely for general input and Ainan Geng for help
with the $^1$H inhomogeneity measurements.

**Financial Support.** This work was supported by the US National Institutes of
Health (R01GM089846) Grants to H.M.A.





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
