# Peer review of "Rapid assessment of Watson-Crick to Hoogsteen exchange in unlabeled"

_Magnetic Resonance, 2021_

## Author Response (AR1)

**Reviewer 1:**

In the manuscript by Liu et al, the authors evaluate if imino 1H experiments CEST can be carried out on unlabeled nucleic acid sample to detect conformational exchange. Although 13C and 15N relaxation dispersion experiments are routinely used to study conformational exchange process like Watson-Crick to Hoogsteen basepair transitions in nucleic acids they require expensive samples enriched in 15N and/or 13C that are also very laborious to prepare and measurements are often restricted to a few judiciously chosen constructs. Hence to study exchange in a large number of sequences to identify sequence dependent conformational dynamics it will be useful to have an NMR experiment that can be used to study exchange in unlabeled sample. The authors show that despite the presence of 1H-1H NOE effects (relatively) 'artifact free' CEST profiles can be obtained by using selective imino excitation and short CEST delays (< 100ms) in unlabeled nucleic acid samples that are significantly cheaper and easier to produce opening up the possibility of studying conformational exchange in several DNA and RNA sequences. I have only a few minor comments.

Here conclusions regarding the Watson-Crick to Hoogsteen basepair transition are being drawn based on a single imino  $\Delta \omega$  value. Is it safe to do this, as breaking of the hydrogen bond will result in  $\Delta \omega$  value of ~-1.5 ppm similar to the  $\Delta \omega$  values being observed here. There should a discussion on how robust this conclusion is and if other measurements like pH dependence of the population etc are required to confirm this.

We thank the reviewer for this comment. For most of the data presented in the manuscript (37 1H CEST profiles), the conclusion that the 1H CEST profiles reflect Watson-Crick to Hoogsteen exchange was not only based on  $\Delta \omega \sim$ -1.5 ppm, but also on the agreement between the fitted  $k_{ex}$  and  $p_{ES}$  values with counterparts measured for the Watson-Crick to Hoogsteen exchange using off-resonance 13C / 15N  $R_{1\rho}$  measurements. Nevertheless, the reviewer does raise an important point, which is that for a new application on a new DNA motif, it may be important to verify the nature of the transient state with the use of additional 13C/15N probes or through pH dependent measurements as suggested by the reviewer. We want to stress that we see the utility of the 1H CEST experiment as an initial screening application to find motifs with outlier behaviors or interesting trends which could then be confirmed through additional experiments. To clarify this important point, we made a number of changes throughout the manuscript.

We have modified the wording in the title and throughout the paper (see page 13 and 39) by changing "measuring the Watson-Crick to Hoogsteen exchange" to "assessing the Watson-Crick to Hoogsteen exchange" to emphasize the utility of the 1H CEST experiment as a facile means to initially assess Watson-Crick to Hoogsteen exchange.

In the abstract, we now note,

"The 1H CEST experiment provides a basis for rapidly screening Hoogsteen breathing in duplex DNA, enabling identification of unusual motifs for more in-depth characterization."

We made the following changes to the introduction on page 9:

"It is therefore desirable to have more facile means to initially assess Watson-Crick to Hoogsteen exchange, and to follow up with in-depth characterization for those motifs exhibiting interesting and unusual behavior. For such an initial screening application, we turned our attention to the imino 1H as a probe of the Watson-Crick to Hoogsteen exchange in unlabeled DNA samples."

We also added the following statement on page 13:

"Since no other ESs have been detected to date in several NMR studies of unmodified canonical DNA duplexes (Nikolova et al., 2011; Alvey et al., 2014; Shi et al., 2018; Ben Imeddourene et al., 2020), a single imino 1H probe could be sufficient to reliably map and characterize the Watson-Crick to Hoogsteen exchange."

And on page 32:

"To test the accuracy of the exchange parameters obtained using 1H CEST, we compared the exchange parameters  $p_{ES}$  and  $k_{ex}$ , derived from a 2-state fit of the data to values determined previously using off-resonance 13C and/or 15N  $R_{1p}$ ."

"This comparison also allowed us to further verify that the exchange process detected by 1H CEST does indeed correspond Watson-Crick to Hoogsteen exchange, and to also further assess for potential contributions from NOE effects, which might cause deviations from agreement."

Figure S9: Please specify what is being plotted on the Y axis.

We thank the reviewer for this suggestion. We now describe what is being plotted in the legend of Figure S9: "the values shown are calculated as  $k_1/k_{1,min}$  or  $k_{-1}/k_{-1,min}$ , in which  $k_{1,min}$  and  $k_{-1,min}$  are the smallest  $k_1$  and  $k_{-1}$  values respectively" to the figure legend.

Figure S5: Increase the range of  $\Delta \omega$  values for T9-H3.

We thank the reviewer for pointing this out and have expanded the range of  $\Delta \omega$  values for T9-H3 in Figure S5.

In materials and methods please specify the number of scans and the d1 used to record the 1H CEST data.

We thank the reviewer for this suggestion and have added the statement in the method section on page 47:

"16 scans were used for A6-DNA (1.0 mM) at 5°C, 10°C, 20°C, 25°C, 30°C, and A2-DNA (1.0 mM) at 25°C. 32 scans were used for A6-RNA (0.5 mM) at 25°C. 64 scans were used for A5-DNA (0.2 mM) at 25°C and for A6-DNA (1.0 mM) at 45°C.

We have also specified the d1 in the figure legend 1b.

" $\tau = \frac{1}{2} d_1 = 0.7 \text{ s}$ "

In figure 1b, it might be safe to destroy all the magnetization after the acquisition, to avoid any accidental offset dependence of the starting 1H magnetization.

The g1 gradient (see Figure 1b) is already included to destroy transverse 1H magnetization prior to the initial Eburp pulse.

We have added the following to the method section on page 46:

"The g1 gradient (Fig. 1b) destroys transverse 1H magnetization prior to excitation of imino resonances. This helps to avoid any accidental offset dependence of the starting 1H magnetization"

In the legend to figure 1b specify the 1H carrier is position at various points in the experiment.

We thank the reviewer for this suggestion and have included the following to the legend of figure 1b:

"The 1H carrier is placed far offset (100,000 Hz) during the two heat compensation periods, then moved to the center of the imino resonances prior to the first pulse **a**. Next, the carrier is placed to a specified offset prior to the relaxation delay ( $T_{EX}$ ), then placed back to the center of the imino resonances following  $T_{EX}$ . Finally, it is placed on-resonance with water for water suppression prior to pulse **b**."

In the legend to figure 1b specify the range (in ppm) that is being excited by the Eburp pulse.

We have specified the excitation range of the Eburp pulse in the figure legend:

"Pulse **a** is a 90° Eburp2.1000 shape pulse (typically 3-4 ms) for selective excitation (excitation bandwidth ~2-3 ppm) of imino protons"

Line 346: "However, since no NOE dips were observable for non-imino protons within 2.8 Å (Fig. 3a), a sizeable cross-relaxation contribution from neighboring imino protons is unlikely considering they are separated by a longer internuclear distance of ~3.7- 3.9 Å (Fig. 3a)" This is a bit confusing: In figure 3b, there are NOE contributions in the 0.1s CEST profiles of G2-H1 due to A3-H2 (3.9 Å) when the selective pulse is turned off. This suggests that NOE effects due to T22-H3 (3.9 Å) will be there in 0.1s CEST profile with selective excitation so long as all the iminos are excited by the Eburp pulse. Artefacts might have been reduced because the T22 imino proton exchanges with water or because the artefacts are very close to the G2-H1 dip. However one may get around the problem by exciting just the Guanine nucleotides with the Eburp or by exciting just G2-H1 and not T22-H3 with the Eburp.

The reviewer makes an important point. The reviewer rightly points out imino-imino NOE effects will not be suppressed when all iminos are excited. We deleted the sentence referred to by the reviewer and added the following statement on page 25-26:

"No NOE dips were observed at the chemical shift of imino protons belonging to neighboring residues in 1H CEST profiles measured in DNA and RNA duplexes, and none of the 1H CEST profiles collected in thus study yielded an ES with  $\Delta\omega$  compatible with the imino 1H chemical shift of a neighboring residue. Nevertheless, these NOE effects could be more difficult to assess given that they would be buried within the major dip. While imino-imino 1H NOEs are not suppressed by selective excitation, their contribution is expected to be smaller relative to other NOE dips observed when using non-selective excitation (distances ~2.4 – 2.8 Å between guanosine/thymine imino and cytosine amino/adenine H2) due the larger distance of separation between neighboring imino protons (~3.5 – 3.9 Å) (Fig. 3a)."

We did not observe the G2-H1-T22-H3 (3.9 Å) NOE dip at the expected chemical shift of ~0.8 ppm in the profiles when using short relaxation delays and selective excitation. The 1H CEST profile is symmetric indicative of no exchange. It is difficult to assess whether the NOE dip due to G2-H1-A3-H2 (3.9 Å) is observable when the selective pulse is turned off as the dip could be masked by the dominant NOE dip corresponding to the G2-H1-C23-H4a (2.4 Å) NOE (Figure 3b). It could be that like G2-H1-T22-H3 (3.9 Å), the G2-H1-A3-H2 (3.9 Å) NOE is also negligible due to the longer distance of separation relative to G2-H1-C23-H4a amino.

To further assess the imino-imino NOE effect, we followed the reviewer's suggestion and performed an experiment selectively exciting G10-H1 and G2-H1 in A6-DNA without exciting the imino resonances belonging to either of their two immediate neighbors. Selective excitation of individual imino protons resulted in 1H CEST profiles (Fig. S2) and fitted exchange parameters (Table S1) for G10-H1 and G2-H1 that are within error of those obtained when exciting all imino protons, again indicating that any imino-imino NOE contribution is negligible. This is also supported by the good agreement seen between the exchange parameters obtained using 1H CEST and 13C/15N *R*1p.

We included these new results on page 27:

"To further assess the impact of imino-imino 1H NOEs on the 1H CEST profiles, we examined whether selective excitation of imino protons but not their immediate neighbors results in different 1H CEST profiles relative to an experiment in which all imino protons are excited. We performed an experiment selectively exciting G10-H1 and G2-H1 in A6-DNA without exciting the imino resonances belonging to either of their two immediate neighbors. Selective excitation of individual imino protons resulted in 1H CEST profiles (Fig. S2) and fitted parameters (Table S1) for G10-H1 and G2-H1 that are within error to those obtained when exciting all imino protons, again indicating that any imino-imino NOE contribution is negligible. Finally, the impact of imino-imino NOEs on the determination of the exchange parameters was also be assessed (*vida infra*) through comparison of the exchange parameters derived from fitting the imino 1H CEST profiles with those measured independently using off-resonance 13C and 15N *R*1p RD measurements."

While it is clear that selective imino excitation coupled with short exchange delays (<0.1s) results in imino 1H CEST profiles that are largely free of NOE induced artefacts due to non imino protons, they can still contain artefacts due to imino protons. Hence the authors should include a few guidelines on safely interpreting the 1H CEST data. When can we get  $\Delta \omega$  values, when can we get exchange parameters etc? When do we have to discard the CEST profiles entirely? While the manuscript contains the guidelines in various places summarizing them in a single paragraph will be useful.

We thank the reviewer for this suggestion. We have expanded the paragraph on page 40 to emphasize the importance of analyzing potential NOE effects.

"Our results indicate that NOE effects from cross-relaxation between imino and nonimino protons can be effectively suppressed for DNA and RNA duplexes in the 1H CEST experiments through selective excitation provided that the relaxation delays are short on the order of 100 ms (Fig. 3b). However, care should be exercised to assess iminoimino NOE effects (Fig. 3b), which may also be more substantial for certain noncanonical motifs. Data should be discarded if the ES chemical shifts match those of nearby imino protons identified using 2D [1H, 1H] NOESY experiments or if the magnitude of the dip varies substantially with or without selective excitation, as this could be an indication of NOE effects involving imino and non-imino protons. Finally, we recommend independent verification of the exchange parameters with the use of other methods such as 13C and 15N experiments for motifs exhibiting highly unusual exchange parameters or ES 1H chemical shifts, and this can also help to confirm Hoogsteen bps as the ES."

**Reviewer 2:**

The article entitled « Rapid measurement of Watson-Crick to Hoogsteen exchange in unlabeled DNA duplexes using high-power SELOPE amino 1H CEST « submitted by Liu et al. is an important contribution to the study of Watson-Crick - Hoogsteen exchange occurring in DNA duplexes. The main achievement of the study is the application of a recently found new pulse sequence SELOPE (Schlagniweit et al., 2018) by the group of K.Petzold to DNA duplexes and WC-HG equilibrium. The work is also of a methodological nature with a systematic study of the possible artifacts related to the 1H-1H cross-correlation in the study of systems in equilibrium exchange. There is really a huge wealth of data that are very convincing and that support the main message of the work that is the interest of the application of SELOPE sequence to unlabelled DNA duplexes permitting to obtain large quantity of data about HG-WC equilibrium at a lower cost and with improved efficiency, additionally the method permits to characterize HG-WC equilibrium with a lower HG population and faster kinetics that it was possible using the previously used R1p Relaxation dispersion 13C/15N methodology. The method, because it permit to obtain rather easily a lot of data on many DNA base pairs, is well adapted to the study of WC-HG equilibria whose the dependence from sequence is quite complex.

We have just some comments about specific points

Legend of Figure 1 the delay d1 is not explicitly defined

We thank the reviewer for this suggestion and have specified the d1 in the figure legend

**1b: " $\tau = \frac{1}{2} d_1 = 0.7 s$ "**

To be clearer , we suggest to make mention earlier in the text that the minor shoulder stated in I 297-298 is indicated with the black line ES in the figure 3B

We have modified the text on page 21 to incorporate this good suggestion:

"On the other hand, a minor shoulder was observed in the 1H CEST profile of T5-H3 (Fig. 3b, the  $\Delta \omega$  is highlighted by dashed red lines in the profile and labeled "ES")."

There is no comments on the large variations observed in  $r\chi^2$  in figure 4 (and also Fig S2, S3) by example why so large variations between U9H3 (498-476) and U5H3 (7.2-6.9) with or without exchange while the experimental data shown with the fit appear similar, additionally for proton T9-H3 or G10H1 while a very significant reduction in reduced  $r\chi^2$  is observed when considering or not the existence of WC-HG exchange, justifying clearly the existence of exchange processes for these protons, the reduced  $r\chi^2$  for the correct model (with exchange) remain rather elevated considering what is expected generally (Rangadurai et al. Prog. In Nuclear Magn .Res., 2019). If this results from a peculiar definition of the reduced chideux, the error bars or any other reason is not clear and needs some explanations.

We thank the reviewer for pointing this out. The different  $r\chi^2$  values for different 1H CEST profiles is most likely due to differences in the quality of the NMR data and differences in uncertainty as well as poor estimation of the real experimental uncertainty. Large variations in  $r\chi^2$  were also observed for 13C/15N CEST profiles reported previously (Shi et al., 2019; Liu et al., 2020; Zhao et al., 2014). In general, the  $r\chi^2$  values for  $R_{1\rho}$  profiles (estimated using Monte Carlo simulations) (Rangadurai et al. Prog. In Nuclear Magn .Res., 2019) are smaller than those of CEST likely due to better estimation of the experimental uncertainties. In the case for U9-H3 and U5-H3 1H CEST profiles, the errors for U9-H3 are 0.00239 and 0.00045 for spin lock power 250 Hz and 500 Hz respectively, while these two errors for U5-H3 are 0.00240 and 0.00845 respectively. The ~18-fold differences in the error of 500 Hz data likely contributes to the large difference in  $r\chi^2$ . We have added a statement in the method section on page 49 to explain the variations in  $r\chi^2$ :

"Note that the variations in  $r\chi^2$  values for different 1H CEST profiles in Fig. 4 and Fig. S2-4 are most likely due to differences in the quality of the NMR data, variations of errors and poor estimation of the real experimental uncertainty."

**Lines 374 not clear -6ppm is repeated**

We thank the reviewer for pointing this out. We meant the range of offset is from -6 ppm to 6ppm. To avoid confusion, we have modified the text on page 27:

"we restricted the offset to -6 ppm to 6 ppm when analyzing and fitting the 1H CEST profiles."

**References:**

Alvey, H. S., Gottardo, F. L., Nikolova, E. N., and Al-Hashimi, H. M.: Widespread transient Hoogsteen base pairs in canonical duplex DNA with variable energetics, Nat Commun, 5, 4786, 10.1038/ncomms5786, 2014.

Ben Imeddourene, A., Zargarian, L., Buckle, M., Hartmann, B., and Mauffret, O.: Slow motions in A.T rich DNA sequence, Sci Rep, 10, 19005, 10.1038/s41598-020-75645-x, 2020.

Liu, B., Shi, H., Rangadurai, A., Nussbaumer, F., Chu, C. C., Erharter, K., Case, D. A., Kreutz, C., and Al-Hashimi, H. M.: A quantitative model predicts how m6A reshapes the kinetic landscape of nucleic acid hybridization and conformational transitions, bioRxiv, 2020.

Nikolova, E. N., Kim, E., Wise, A. A., O'Brien, P. J., Andricioaei, I., and Al-Hashimi, H. M.: Transient Hoogsteen base pairs in canonical duplex DNA, Nature, 470, 498-502, 10.1038/nature09775, 2011.

Shi, H., Clay, M. C., Rangadurai, A., Sathyamoorthy, B., Case, D. A., and Al-Hashimi, H. M.: Atomic structures of excited state A-T Hoogsteen base pairs in duplex DNA by combining NMR relaxation dispersion, mutagenesis, and chemical shift calculations, J Biomol NMR, 70, 229-244, 10.1007/s10858-018-0177-2, 2018.

Shi, H., Liu, B., Nussbaumer, F., Rangadurai, A., Kreutz, C., and Al-Hashimi, H. M.: NMR Chemical Exchange Measurements Reveal That N(6)-Methyladenosine Slows RNA Annealing, J Am Chem Soc, 141, 19988-19993, 10.1021/jacs.9b10939, 2019. Zhao, B., Hansen, A. L., and Zhang, Q.: Characterizing slow chemical exchange in nucleic acids by carbon CEST and low spin-lock field R(1rho) NMR spectroscopy, J Am Chem Soc, 136, 20-23, 10.1021/ja409835y, 2014.